Digging the pupfish out of its hole: risk analyses to guide harvest of Devils Hole pupfish for captive breeding

Beissinger Steven R. beis@berkeley.edu
Department of Environmental Science, Policy & Management, and Museum of Vertebrate Zoology, University of California , Berkeley, CA , USA
Roberts David
Electronic publication date: 2014 Sep 9
Publication date: 2014
Volume: 2
Electronic Location ID: e549
Received 2014 Jun 16; Accepted 2014 Aug 7
Copyright: © 2014 Beissinger
Copyright year: 2014
Copyright holder: Beissinger
License: This is an open access article distributed under the terms of the Creative Commons Attribution License, which permits unrestricted use, distribution, reproduction and adaptation in any medium and for any purpose provided that it is properly attributed. For attribution, the original author(s), title, publication source (PeerJ) and either DOI or URL of the article must be cited.
License URL: https://creativecommons.org/licenses/by/4.0/

Keywords: Population viability, Captive breeding, Devils Hole pupfish, Harvest strategy, Extinction

Funding: This work was funded by a contract from the National Park Service Portions of this work were funded by the National Park Service. The funder had no role in study design, data collection and analysis, decision to publish, or preparation of the manuscript.

==============================
The Devils Hole pupfish is restricted to one wild population in a single aquifer-fed thermal pool in the Desert National Wildlife Refuge Complex. Since 1995 the pupfish has been in a nearly steady decline, where it was perched on the brink of extinction at 35–68 fish in 2013. A major strategy for conserving the pupfish has been the establishment of additional captive or “refuge” populations, but all ended in failure. In 2013 a new captive propagation facility designed specifically to breed pupfish was opened. I examine how a captive population can be initiated by removing fish from the wild without unduly accelerating extinction risk for the pupfish in Devils Hole. I construct a count-based PVA model, parameterized from estimates of the intrinsic rate of increase and its variance using counts in spring and fall from 1995–2013, to produce the first risk assessment for the pupfish. Median time to extinction was 26 and 27 years from spring and fall counts, respectively, and the probability of extinction in 20 years was 26–33%. Removing individuals in the fall had less risk to the wild population than harvest in spring. For both spring and fall harvest, risk increased rapidly when levels exceeded six adult pupfish per year for three years. Extinction risk was unaffected by the apportionment of total harvest among years. A demographic model was used to examine how removal of different stage classes affects the dynamics of the wild population based on reproductive value (RV) and elasticity. Removing eggs had the least impact on the pupfish in Devils Hole; RV of an adult was roughly 25 times that of an egg. To evaluate when it might be prudent to remove all pupfish from Devils Hole for captive breeding, I used the count-based model to examine how extinction risk related to pupfish population size. Risk accelerated when initial populations were less than 30 individuals. Results are discussed in relation to the challenges facing pupfish recovery compared to management of other highly endangered species.

Introduction

The Devils Hole pupfish (Cyprinodon diabolis) may have the smallest geographic range of any vertebrate in the wild. This species is restricted to a single population and occurs primarily in the upper 10 m of Devils Hole, an aquifer-fed thermal pool (∼33.5 °C) and limestone cavern with a surface area of 50 m2 that is located 17 m below the land surface in the Desert National Wildlife Refuge Complex in Nye County, Nevada (Andersen & Deacon, 2001; Baugh & Deacon, 1983a). Pupfish spawning takes place predominately on a shallow (∼0.35 m), submerged ∼2 × 4 m shelf (Hausner et al., 2013; James, 1969). It is the only fish in Devils Hole where it has resided since climate warming caused regional drying beginning ∼20,000 years ago (Szabo et al., 1994).

The Devils Hole pupfish (DHP) has played an important role in the history of the conservation movement. In 1952 Carl Hubbs, a pioneer of western ichthyology, convinced President Harry Truman to designate Devils Hole as a disjunct part of Death Valley National Monument to protect both its unique geological features and the pupfish (Riggs & Deacon, 2004). The DHP was among the initial species listed when the U.S. Endangered Species Preservation Act was passed in 1967. A monitoring program was begun that used scuba divers to count fish and initial counts found 100–250 individuals in the early 1970s (Riggs & Deacon, 2004). Soon after, the DHP was the subject of a historic four-year legal battle over water rights that culminated in the U.S. Supreme Court (1976 Cappert v. United States). The decision caused a cessation of groundwater mining from further dewatering the aquifer for development in the Amargosa Valley and nearby Ash Meadows (Riggs & Deacon, 2004), and the case had a major influence on ground water rights in the region.

With the cessation of groundwater pumping, water levels rose in Devils Hole as did DHP numbers (Andersen & Deacon, 2001), which reached a maximum count of 541–548 individuals in 1980, 1990 and 1995 (Fig. 1). Since 1995, however, the pupfish in Devils Hole has been in a nearly steady decline, where it was perched on the brink of extinction at 35–68 fish in 2013. Hypothesized causes of decline include food limitation due to changes in the Devils Hole algal and invertebrate communities (KP Wilson, pers. comm., 2014), mutational meltdown caused by centuries of reduced genetic variation (Martin et al., 2012), and climate warming resulting in increased water temperatures and decreased dissolved oxygen (Hausner et al., 2013; Hillyard et al., 2014). The population decline was amplified by the accidental deaths of at least 72 pupfish in Sept. 2004 in larval traps that were washed into Devils Hole during a flash flood (Manning & Wullschleger, 2004).

Figure 1 Maximum spring and fall counts of the Devils Hole pupfish.

A major strategy for conserving the DHP, in addition to securing Devils Hole, has been the establishment of captive or “refuge” populations (Baugh & Deacon, 1988; Wilcox & Martin, 2006). However, past attempts have not been successful. Between 1969 and 2012, multiple attempts were made to propagate the DHP in captivity and establish refuges. Typically these efforts removed 12–30 pupfish at a time from Devils Hole. They were transported to 14 different locations ranging from managed efforts in established, commercial aquaria to constructed and natural ponds or springs with less oversight. All ended in failure, with some populations lasting for 1–2 decades and others for 1–2 years (Karam, 2005; Wilcox & Martin, 2006). Reasons for failure, when known, were varied and included equipment or water supply failure, predation by native and exotic species, vandalism, failure to reproduce, and hybridization.

Now in response to the recent population collapse and imminent threat of extinction, a new captive propagation facility has been built and, unlike previous efforts, was designed specifically to breed pupfish. It was built in nearby Ash Meadows, and received a total of 60 DHP eggs from August 2013 to January 2014. The Ash Meadows facility provides a new opportunity for successful captive breeding, but it also presents a challenge. Without an obvious track record of captive breeding success, can a captive population be initiated without unduly accelerating the risk of extinction of the DHP in Devils Hole? Which life stages of the fish should be collected to minimize impact on the wild population? On the other hand, if the trajectory of decline for the DHP continues, will there be a moment when all individuals should be removed from the wild in order to maximize the genetic diversity for captive breeding to succeed? This occurred for the California condor (Gymnogyps californianus) in 1986 when evidence indicated little hope of survival in the wild (Snyder & Snyder, 1989). That controversial and difficult decision was implemented after much debate within the U.S. Fish and Wildlife Service over whether to proceed with removing the last wild condors, and after litigation by its conservation partner, the National Audubon Society, to prevent it. Removing the last wild individuals turned out to be the right decision, as it secured a gene pool for the subsequently highly successful captive propagation program (Snyder & Snyder, 2000), although reintroduction to the wild remains problematic (Walters et al., 2010). Successful captive breeding was foreseen for condors based on captive propagation undertaken with surrogate species, such as the Andean condor (Vultur gryphus), and similar possibilities exist for the pupfish based on experiences with the DHP and its congenerics (e.g., Baugh & Deacon, 1983b; Deacon, Taylor & Pedretti, 1995; Lema & Nevitt, 2006).

Here I produce the first risk assessment for the Devils Hole Pupfish. I construct a count-based population viability model to project the risk of extinction for the DHP parameterized from recent biannual pupfish surveys. I then use the model to evaluate how the number of individuals removed for captive breeding and the timing of harvest affects population viability of the DHP in Devils Hole. Next I build a matrix population model to ask what the impact of removing pupfish eggs, early life stage (larval) individuals, or adults will be on DHP population dynamics based on patchy demographic data, expert opinion and data borrowed from closely-related species. Finally, I return to the count-based population model to ask how extinction risk relates to DHP population size to evaluate when it might be prudent to remove the remaining pupfish from Devils Hole for captive breeding. In these applications of population models to project extinction risk, commonly called population viability analysis or PVA (Beissinger & McCullough, 2002), I evaluate conservation decisions by comparing differences in projected outcomes among management options incorporated into the models rather than basing recommendations solely on the projected rates of extinction (Lotts, Waite & Vucetich, 2004; McCarthy, Andelman & Possingham, 2003; Ralls, Beissinger & Cochrane, 2002; Reed et al., 2002).

Materials and Methods

Modeling risk and harvest strategies for captive breeding using DHP counts

Visual counts of pupfish from 1972 to 2013, made by SCUBA divers in the pool and by observers on the shallow shelf, were obtained from Death Valley National Park personnel (Fig. 1). See Dzul et al. (2012) for survey details and sources of error. Counts of pupfish did not distinguish between adults and detectable early life stage individuals, and were conducted on about a monthly basis from 1972–1983. By 1985 counts were mainly done biannually, in the spring (March–April) during the main breeding season of the pupfish and in the fall (Sept.–Oct.) during a second but reduced pulse of breeding. Only one count per season was conducted in most years, which precluded the use of N-mixture models of population estimation that require repeated sampling (Royle & Dorazio, 2008). In the absence of survey-level covariates of effort, I was unable to use a single visit conditional likelihood approach to estimate abundance (Lele, Moreno & Bayne, 2012; Solymos, Lele & Bayne, 2012). Thus, I choose to use the maximum count for each season in each year as an estimate of the population size, creating two time series of counts. Treating counts separately provided independently derived estimates of population trends and permitted seasonal evaluation of harvest options.

I calculated the rate of population growth (r = ln(Nt+1/Nt)) from pairs of counts of population size (N) from consecutive years (t). I then fit one density-independent and two density-dependent models of population growth to the period when pupfish numbers grew (1972–1995) and the period of population decline (1996–2013) following Morris & Doak (2002):

Exponential (density-independent): (1) lnNt+1/Nt=r+εi

Logistic (density-dependent): (2) lnNt+1/Nt=r1−NtK+εi

Theta-logistic (density-dependent): (3) lnNt+1/Nt=r1−NtKθ+εi

where r = growth rate, K = carrying capacity, θ adjusts how population growth changes with N, and εi = is the variance or deviation in the natural logarithm of population growth (lnr) centered around zero. Akaike’s Information Criterion (AIC) corrected for small sample size (AICc) was used to quantify model fit (Burnham & Anderson, 2002).

To model extinction risk in the wild, the pupfish population in Devils Hole was projected forward in time for 100 years with separate models for spring and fall using each season’s fitted values for average population growth rate, carrying capacity (when appropriate), and annual deviations from mean growth rates (εi) estimated from 1996–2013 to yield the median time (years) to extinction and probability of extinction. Starting population size was set to the number of pupfish counted in 2013 for spring (35) and fall (68), and 10,000 iterations were run. Fractional numbers of individuals were rounded down each year. Projections were done using the program @Risk (@Risk 6.2; Palisade Software, Ithaca, New York) in Microsoft Excel, as were all simulations presented below.

Using εi to model annual variation among counts has contrasting effects on the resulting estimate of extinction risk. The εi term incorporates implicit effects captured in DHP count fluctuations including: (1) demographic, environmental, and genetic stochasticity; (2) catastrophes; and (3) sampling variation. Nevertheless, these processes were not explicitly modelled. The inclusion of sampling variation in the estimate of εi will overestimate variation in population growth and inflate extinction risk. However, in the absence of explicit incorporation in the model of genetic processes (e.g., inbreeding depression), extinction risk may be underestimated. As I am unable to determine the magnitude of each effect, estimates of extinction risk are best interpreted when compared among different scenarios.

To evaluate the effects of different strategies for removing individuals to initiate a captive breeding program on the wild population, I used the models that best described DHP population dynamics for spring and fall from 1996 to 2013, and harvested (removed) different numbers of individuals (0–14) at the start of each simulated year. Harvest was done for each of three years to mimic building a new population for captive propagation. Median time to extinction and probability of extinction were evaluated from 10,000 iterations.

I used the same model to examine how risk of extinction changes with DHP population size to evaluate when to remove all pupfish individuals from Devils Hole based solely on changes in wild population risk without considering genetic goals for the captive population. The stochastic count-based PVA model was run for 10,000 iterations parameterized with the 1996–2013 measures of population growth, incrementally changing the initial population size.

Deterministic matrix demographic model to evaluate age classes to remove

A deterministic demographic model was developed to evaluate the effects of removing individuals of different life stages on DHP population dynamics by calculating for each life stage its reproductive value (i.e., expected future number of offspring produced) and elasticity (sensitivity of population growth to changes in demographic parameters associated with each stage). I sought opinions for constructing and parameterizing the model from 15 DHP and fisheries experts that attended the DHP Risk Analysis Workshop (8 Nov. 2013). Demographic data available for the DHP are so limited, and the uncertainties so large, that I could not justify the choice of particular rates or scenarios. Instead, I generated 5,000 matrices composed of random combinations of potential average demographic rates for the pupfish chosen from uniform distributions that sampled means between their possible minimum and maximum values (Fig. 2). This approach allowed exploration of the potential parameter space, resulting distribution of reproductive value and elasticity for stage classes, and relationships among them.

Figure 2 Devils Hole pupfish postbreeding life cycle, projection matrix, and range of weekly demographic rates used in creating matrices for analysis.

A post-breeding projection was chosen to enable inclusion of eggs as a life stage because they are potential targets for management. It was based on a life cycle diagram and projection matrix with 3 stages (Fig. 2): eggs, early life stage or ELS (from hatching at 4 mm to 11 mm in length) and adults (>12 mm). A 7-day time step was used based on the time required for an egg to hatch and become an ELS. I converted DHP demographic rates expressed in the literature on a monthly basis to a daily rate (divided by 30) and then to a weekly rate (multiplied by 7).

The model required estimates for the proportion of individuals that: (a) survive over a time step and remain Adults (P3); (b) grow from Eggs and survive to become ELS (G1); and (c) grow from ELS to become Adults (G2), or survive and remain as ELS (P2). It also required estimates for realized fecundity, which is the product of (d) the proportion of individuals that survive and remain (P3 for Adults) or that grow to a reproductive stage over the time step (G2 for ELS), and (e) their fecundity (m3). Because male fecundity was not known, rates were based on females and sex ratio of eggs was assumed to be 50:50.

As the overall probability of survival (S) for a stage class is S = P + G, I estimated the proportion of individuals surviving and transitioning using the equation: (4) G=St1+S+S2+S3⋯St−1

where t = number of time steps in a stage including the transition to the next stage. Estimating G2 in this manner reduced the number of demographic estimates the model required to four (P3, S2, G1, and m3), but required an estimate of the number of time steps a newly arrived ELS (4 mm) needed to grow to be an adult (>11 mm). James (1969, p. 46) estimated growth rates of caged individuals of 4.7 mm per month for offspring in Devils Hole, which would require 1.6–2 months (7–8 weeks) for growth from ELS to Adult. Hybrid DHP, which may grow faster than nonhybrids, require 30–45 days to grow from hatching to adult (O Feuerbacher, pers. comm., 2013). I let the number of weeks (time steps) that ELS require to reach adult size range from 4 to 7.

Monthly survival rates in the literature are based on best professional judgment, as a DHP capture–recapture study has not been conducted in Devils Hole. Adult survival (P3) in Dzul et al.’s (2013) model was assigned values of 0.70, 0.80, and 0.86, while Chernoff (1985) used 0.91. I explored values ranging from 0.7 to 0.9. Monthly survival (S2) assigned by Dzul et al. (2013) to juveniles (ELS in my model) was 0.047, 0.072 and 0.097, based primarily on simulation runs of their model that resulted in the stable population growth observed during their one-year study. I allowed ELS monthly survival to range between 0.05 and 0.15.

Fecundity (m3) has not been measured for DHP in the wild. Females are thought to lay a single egg with each spawn, which can occur at any time of year but peaks from mid-February to mid-May with a secondary peak from July–Sept (Hausner et al., 2013; Lyons, 2005). Fecundity is unrelated to size in the DHP (Minckley & Deacon, 1973; Mire & Millett, 1994; Shrode & Gerking, 1977). Minckley & Deacon (1973) stated that “an average female may have about 4–5 ova or 10–20% of her total complement in a mature condition during a peak reproductive season”. This suggests that a female might be capable of laying 20–50 eggs during her lifetime (minimum 4 × 5 = 20; maximum: 5 × 10 = 50) and that the eggs would be distributed differentially during the year. Chernoff (1985), using data in James (1969) and Minckley & Deacon (1973), estimated the average number of eggs spawned per female to be 24 and the average per female per month during the breeding season from 1–5. In summary, females may lay 0–4 eggs per week. Assuming a 50:50 sex ratio, I set reproductive rates to range from as low as 0.05 female eggs per week to as many as 2 female eggs per week. In the absence of hatching success measures, I assumed all eggs survived for one week and hatched, which may overemphasize their importance to population dynamics.

Results

Estimation of population growth measures

Fits of density dependent and independent models of population growth (Table 1) reflect the shift in pupfish dynamics over time (Fig. 1). From 1972 to 1995, density dependent models provided the best fit for both spring and fall counts (cumulative AIC weight of 0.99–1.0), with the logistic model fitting best for both seasons (Table 1). Growth was positive before 1996 (r = 0.82; 95% CI [0.40–1.24]), with a carrying capacity of 216 (95% CI [194–237]) for spring and 444 (95% CI [390–498]) for fall. From 1996–2013, density independent models best fit the DHP counts, with a declining population growth (r) of −0.049 (95% CI [−0.24–0.15]) and −0.088 (95% CI [−0.27–0.09]) for spring and fall counts, respectively. Associated values for annual environmental variation were ε = 0.117 for spring and ε = 0.096 for fall.

Table 1 Model selection and model estimates for DHP counts by time period.

Bolded models are the best, based on their AICc score for the years and seasons evaluated. AICc weight (wt) indicates the strength of a model relative to other models in that set of years and season.

Years	Season	Model	ΔAICc	AICc wt	r	K	θ	
1972–1995	Spring	Logistic	0.0	0.84	0.820	215.9		
		Theta-logistic	3.3	0.16	0.691	216.7	1.2	
		Exponential	10.5	0.00	0.029			
	Fall	Logistic	0.0	0.78	0.574	444.0		
		Theta-logistic	2.6	0.22	0.328	452.0	2.1	
		Exponential	9.5	0.01	0.034			
1996–2013	Spring	Exponential	0.0	0.59	−0.049			
		Logistic	1.0	0.36	0.201	110.0		
		Theta-logistic	4.8	0.05	0.676	95.8	0.3	
	Fall	Exponential	0.0	0.64	−0.088			
		Logistic	1.4	0.32	0.136	140.0		
		Theta-logistic	5.4	0.04	0.565	134.0	0.3	

Count-based risk analysis for the DHP in Devils Hole

Time to extinction for the DHP estimated separately from spring and fall counts followed the expected logarithmic distribution, resulting in most simulated populations becoming extinct within 50 years and a smaller number remaining extant for extended periods (Fig. 3A). Median and mean time to extinction were 26 and 27 years, and 17 and 22 years, respectively, for spring and fall counts (Fig. 3A). The chance of extinction within a decade was less than 5%, but rose rapidly to 26–33% by 20 years, ∼45% by 25 years, and 81–90% by 50 years (Fig. 3B).

Figure 3 Risk projections for Devils Hole pupfish based on estimates from Spring and Fall counts: (A) time to extinction; and (B) probability of extinction.

Effect of harvest for captive propagation on extinction risk in Devils Hole

Risk to the wild population of removing pupfish from Devils Hole to initiate captive breeding depended on the level and timing of harvest (Fig. 4). Removing individuals in the fall had less impact on risk to the wild population than the same level of harvest in spring, due to the larger initial size of the pupfish population in fall. For both spring and fall harvest, risk increased linearly with the number of individuals removed at rates of up to 6 pupfish per year for 3 years. Above this level of harvest, extinction risk accelerated; the median time to extinction fell rapidly (Fig. 4A) and the probability of extinction increased at a greater rate (Figs. 4C and 4D), especially for spring harvest.

Figure 4 Pupfish extinction risk in Devils Hole in relation to (A) Harvest season (spring or fall), (B) apportionment of the total number of individuals harvested among 1, 2 or 3 years; and harvest level (0–14 fish per year for 3 years) for spring (C) and fall (D).

Surprisingly, extinction risk was relatively unaffected by the apportionment of the total harvest among years (Fig. 4B). Whether the total number of individuals was removed over one, two or three years had little effect on the probability of extinction at 20 years.

Impact of removing different age classes on DHP population dynamics

Analysis of reproductive value (RV) and elasticity both indicated that Adults were the stage class with the greatest influence on population dynamics (Table 2). RV for ELS (Stage 2) was nearly identical to RV of Eggs (Stage 1) and never exceeded 1.2 in the 5,000 iterations of matrices evaluated. RV for Adults, however, averaged 24.2 ± 0.3 and ranged from a minimum of 4.32 to a maximum of 123 with a median value of 17.7 (Fig. 5A). This pattern occurred partly because so few ELS survived to become adults. Adult RV was strongly negatively related to ELS growth and positively related to the stage duration (Fig. 5B). Elasticity results also indicated that changes in adult survival (P3) had by far the greatest influence on population growth rates (Table 2).

Figure 5 Reproductive Value (RV).

(A) Distribution of the Reproductive Value (RV) for Adults (stage 3) scaled relative to Eggs (stage 1). (B) Scatterplots of Adult RV (scaled) vs. demographic rates used a input for the matrix model, color coded by ELS stage duration or the number of weeks required to reach adult size (4, blue, 5, purple, 6, black and 7, green).

Table 2 Reproductive values (±SE) scaled relative to Eggs (stage 1) and elasticity from analyses of 5,000 potential demographic matrices of the Devils Hole pupfish.

See Fig. 2 for life cycle diagram, matrix construction and rates.

		Elasticity	
	Reproductive value	Fecundity	Growth	Survival	
Life stage	(RV)	(Gm or Pm)	(G)	(P)	
Egg (1)	1.00	0	0.049	0	
ELS (2)	1.01 ± 0.01	0.004	0.056	0.045	
Adult (3)	24.4 ± 0.29	0.045	0	0.800	

Extinction risk vs. DHP population size in Devils Hole

As expected, the median time to extinction increased and probability of extinction in 10 years decreased when simulations were begun with larger initial population sizes (Fig. 6). Both metrics of extinction risk changed linearly with initial population sizes between 30 and 50 individuals, but accelerated when initial populations were less than 30 individuals.

Figure 6 Extinction risk vs. population size.

Extinction risk (circles, median time to extinction; squares, probability of extinction in 10 years) vs. population size at the start of the simulation for spring (blue) and fall (red).

Discussion

The Devils Hole Pupfish has been threatened with extinction since the Endangered Species Act was enacted. A count-based PVA suggests this population faces a 28–32% chance of going extinct in the wild over the next 20 years (Fig. 3), but could disappear within 7 years. This analysis undoubtedly underestimates extinction risk, as my model does not explicitly incorporate the effects of inbreeding, demographic stochasticity, and catastrophes (Mangel & Tier, 1994; Ralls, Beissinger & Cochrane, 2002), depending instead on their intrinsic influences being accounted for in the estimates of population growth and its variances derived from pupfish counts. Nevertheless, the projected risk of extinction is moderately high; for comparison, IUCN red list criteria based on extinction risk from a PVA associates “Critically Endangered” with forecasts of 50% extinction within 10 years and “Endangered” with forecasts of 20% extinction within 20 years (IUCN, 2014). Even when the population was growing or stable from 1972 to 1995, the average long-term carrying capacity in Devils Hole was small (216 in spring and 444 in fall), as estimated from logistic population growth (Table 1). Thus, despite the lack of long-term success with captive propagation and refuge populations in the past, the need to establish a secure population outside Devils Hole is greater than ever if this species is to persist.

Risks to the wild population when building the captive population

Removing individuals from the wild to initiate captive breeding will reduce the time to extinction of pupfish in Devils Hole (Fig. 4). Risks can be mitigated by harvesting wild pupfish in the fall, when the population tends to be larger, rather than in the spring, when less recruitment has occurred. However, risk to the wild population accelerated when more than six adults were harvested annually, although this effect is smaller in the fall. In any case, it may be unwise to harvest adults for captive propagation if eggs or early life-stage individuals are available and can be raised in captivity. Removing eggs has the least impact on the population dynamics of DHP in Devils Hole (Fig. 5, Table 2). The egg stage had the lowest reproductive value and the lowest elasticity, although population dynamics was nearly as insensitive to incremental, instantaneous changes in the rates of early life stage individuals.

Results from the two population models can be connected by translating reproductive value of the egg stage from the demographic matrix model to a number of adults that can be removed in the count-based models. The mean RV of an adult pupfish is roughly 25 times greater than that of a pupfish egg in Devils Hole (Table 2). From this perspective, removing 25 eggs for captive breeding is equivalent to removing a single adult in terms of its influence on population dynamics (Caswell, 2001).

The success of removing pupfish eggs or adults to build the captive population and support the wild population depends upon subsequent husbandry and (re)introduction. This requires success in each of a series of steps (Armstrong & Seddon, 2008; Snyder et al., 1996): (1) survival, growth and reproduction in captivity; (2) maintenance in captivity of genetic diversity and a viable gene pool; and (3) successful preconditioning and release to the wild, either to establish a new refuge in a secure location or to bolster the pupfish in Devils Hole, ideally after conditions that caused the population to decline have been identified and ameliorated. While the first step in the process has been successfully accomplished, as eggs taken from Devils Hole have been hatched in captivity, serious obstacles remain before releases of pupfish can occur. Unfortunately, the longer that pupfish remain in captivity before being returned to the wild, the greater the likelihood of selection for domestication and loss of behaviors needed to survive in the wild (Ford et al., 2008; Frankham, 2008; Kelley, Magurran & Macías García, 2006; Snyder et al., 1996). Avoidance of domestication will be a key priority in managing the captive population (LH Simons, pers. comm., 2014).

When to designate the “condor moment” and remove all pupfish from Devils Hole

Designating the “California Condor moment” for the DHP–when removing the remaining pupfish from Devils Hole would be the best course of action–is a management option conservation biologists have rarely considered. Similar situations have been faced in a few other extreme cases, such as the Hawaiian Crow (Corvus hawaiiensis) which was rescued from extinction but cannot be reintroduced to the wild due to the persistence of the toxoplasmosis responsible for its decline (Work et al., 2000), and the black-footed ferret (Mustela nigripes) which was captured for captive breeding after a few individuals were rediscovered in the 1980’s (Clark, 1990; Jachowski et al., 2011). I evaluated when to designate the condor moment by searching for nonlinearity in the relationship between risk of extinction and population size (Fig. 6). Extinction risk accelerated when population size fell below 30 individuals. However, the risk analysis presented here only partly addresses the issue of when to intervene by removing all individuals from the wild to save the DHP. The degree that genetic diversity of the wild population is represented in the captive population is a major concern that the model does not address.

There are many similarities between the situation currently facing the DHP and the decline of the California Condor in the mid-1980s (Meretsky et al., 2000; Snyder & Snyder, 2000; Snyder & Snyder, 1989). Both species experienced a rapid population decline occurring over decades as a result of poorly understood causes that were difficult to reverse. Neither species had a history of successful captive breeding, although experiences with the target or a surrogate species indicated the potential for success. Neither species was sufficiently represented in captivity to conserve genetic diversity. Concerns were voiced for condors that capturing the remaining individuals for captive breeding would reduce the need to conserve of habitats needed for their reintroduction, and similar issues relating to water rights affect the pupfish. Finally, a long history of management controversies and struggles to conserve each species has made them conservation icons.

Some key differences, however, between pupfish and condor life history and management could make it easier to recover the DHP. First, the “faster” life history of the DHP (early maturation, reproduction and short life span) promotes rapid population growth and recovery compared to the “slower” life history” of the condor (delayed age of first breeding, low level of reproduction and long life span). Yet, it also dictates that immediate reproductive success in captivity must occur, given the one year lifespan of the pupfish. Second, reintroduction of captive-reared DHP to the wild should be much easier than it has been for the condor. Reintroduction of condors has been on-going for two decades without achieving a self-supporting population due to poisoning from ingestion of lead fragments in their food, excessive tameness of released birds, parents feeding microtrash to nestlings, and other causes of mortality (Meretsky et al., 2000; Walters et al., 2010). DHP habitat is protected and threats to the species should be easier to control, at least in theory.

As a result of the potential for fast population growth in captivity, it is not too late to rescue the Devils Hole pupfish from extinction. Needed now to ensure success is a diagnosis of the causes of decline in Devils Hole in order to recover the wild population, an evaluation of the extent that the DHP harbors a significant genetic load and whether this requires a genetic restoration strategy (Martin et al., 2012), an analysis of the risks (e.g., introduction of diseases or parasites) and benefits (e.g., genetic and demographic rescue) of connectivity between the captive population and Devils Hole, and an evaluation of locations for new refugia to introduce the pupfish and better analysis of why refugia failed in the past. Of key importance will be maintaining the wild pupfish population in Devils Hole, while launching the new captive breeding facility.

Supplemental Information

Supplemental Information 1 Counts of Devils Hole pupfish

Click here for additional data file.

This manuscript benefitted greatly from input of the participants at the Devils Hole Pupfish Risk Analysis Workshop (8 Nov. 2013), and from reviews by Christopher Clements, Daniel Gaines, Sean Maher, J. Michael Reed, Lee Simons, Noel F.R. Snyder, Ian Watson, Kevin Wilson, and the Beissinger lab.

Additional Information and Declarations

Competing Interests

Author Contributions

The author declares there are no competing interests.

Steven R. Beissinger conceived and designed the experiments, performed the experiments, analyzed the data, contributed reagents/materials/analysis tools, wrote the paper, prepared figures and/or tables, reviewed drafts of the paper.

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
