# Peer review of "Digging the pupfish out of its hole: risk analyses to guide harvest of Devils Hole pupfish for captive breeding"

_PeerJ, doi:10.7717/peerj.549_

## Round 0.1 · original submission · Minor Revisions

· Academic Editor

Minor Revisions

This is a fascinating paper that requires some clarification in places as outlined by reviewers 1 and 3.

·

Basic reporting

This article is well written, reasonably concise, and covers an interesting topic in a thorough manner.

Experimental design

Some issues need clarification.

– Line 88 – this suggests that counts of adults and early life stage individuals are separate. If this is so, is it reasonable to use population counts that include both adults and juveniles, even though as you say later there is a high mortality rate in juvenile individuals?

– In the introduction it is noted that there have been a number of previous attempts to create stand-alone populations by removing individuals from Devil's hole. The number given is between 12 and 30, which could be a large proportion of the remaining individuals in Devil's hole. Is this is taken into account in your analysis of survival and rates of population growth?

– Table 2 – I suggest a rotating this through 90°, with the life stages of the fish in a column on the left hand side, and the rates etc as set out columns.

– Figure 3 – the X axis in figure 3B would be more easily understandable if it was the actual years – i.e., 2013, 2023, 2033… Part of the axis of this figure are cut off at the edges. fig 3A - Given that simulations are run up to 100 years, giving the maximum valueof extinction as 100 years seems a little counterintuitive. Perhaps you can either run the simulations for a longer period of time into the future, or take out the "maximum" values.

Figure 4 – the arrangement of the panels is unnecessarily confusing. I suggest making it into three rows, panel a on the first, the two panels of panel b on the second, and the panel c on the third.

– Figure 5A – hard to understand. X and Y axes needs to be clearer, especially the X axis. There also appears to be a table of maximum, mean, standard deviation overlaid on the right-hand side of the graph. This does not look intentional.

Validity of the findings

The author has taken a robust approach to the uncertainties surrounding the issues tackled, often by using a range of parameter values, an approach that is welcomed. Some issues need clarification, and the majority of these are covered in this section "experimental design".

Additional comments

In general this is a well-written piece, with a clear message. Some improvements could be made however. Below are listed some points which could be considered:

– In the abstract, it states that when harvest rates exceed six fish per year extinction risk rapidly increases. Is this six adult fish? Juveniles? eggs?

– The first paragraphs of the introduction could be combined and made to flow better. The information presented is interesting and important in the setup of the article however it is currently a little awkward to read.

– Line 216 – "absent hatching success measures" doesn't make sense to me.

Line 243 "likely due to the… In fall", this is discussion material

– Line 281 – URL for IUCN can go in references,

Line 282 – do you mean carrying capacity? I suspect you mean maximum observed number of individuals

– Line 344 to 349 – Lifestyle should be replaced with life-cycle, or life history

–line 283 - different values than those in the introduction

·

Basic reporting

This is quite good enough. The history of the DHP is described in enough detail for the reader to understand the problems faced by those trying to conserve it. It complies with the requirements of PeerJ.

Experimental design

This does seem to cover all the bases. The overall modelling is good enough to identify that the DHP is in real trouble but the addition of key extinction point probabilities is a useful addition which hopefully will concentrate minds. Not being familiar with the PVA model, I cannot comment in detail but I assume it have been validated for this kind of study. The model is run enough times to show that extinction in the wild seems more of a certainty than a likelihood in the not too distant future.

Validity of the findings

Assuming the validity of the model and the data put into it (the numbers of DHP are very low which presumably creates some degree of uncertainty compared to running the model with large populations or multiple populations) but the iterations do all seem to point in one direction, the inevitability of extinction. The findings are valid, but see below.

Additional comments

While not necessarily the role of this paper, someone does need to take the hard decision over whether the DHP can be conserved in the wild and whether it is cost-effective use of funds to do so or just admit it is doomed and decide whether it is practical to conserve it in captive populations or just let it go extinct. Previous efforts at ex-situ conservation have failed and there is a need to analyse this in more detail to see why. Examples from other attempts at transplanting some fish to new sites have shown a very high habitat specificity which can lead to failure to establish a new population. It might even be that the DHP will need to be conserved in captivity and only returned to the wild if ecological conditions at Devil’s Hole improve.

·

Basic reporting

I’m not sure figure 5B would actually help me reproduce the analysis in this paper.
no other comments

Experimental design

The modeling methods are excellent, and do not go beyond available data. I have one suggestion. Is there an easy way to add in the dynamic of putting fish back into Devil’s Hole from the captive population? Making reintroduction dynamic, as removal currently is, would be an important piece for figuring out the optimal management pattern to minimize extinction in the wild. It should also change the answer on ‘when to pull the plug’.

Validity of the findings

no comments

Additional comments

The Devil’s Hole pupfish (DHP) is a famous example of a species persisting in isolation. Its recent severe decline makes it particularly vulnerable to extinction, and desperate measures should certainly be considered for this species’ preservation. Captive breeding is always controversial, as Beissinger points out – although, he identifies CA condors as a success story (line 65), there are those who argue that it is not (at least using the criterion of reintroduced establishment). This paper is an excellent exploration of the potential it has for DHPs. With the new captive breeding facility that was built, it appears that the decision to breed DHP in captivity has been made – so Beissinger’s paper provides much-needed information on critical management questions.

Consequently, I think this paper provides excellent thoughts and guidance on expectations from different captive breeding scenarios, with the following caveat: Since there have been at least 3 or 4 attempts at captive breeding of DHP, all of which failed – while the wild population has persisted – what has changed that makes it a reasonable expectation that this time will succeed?

Also, to accurately predict when to take all fish into captivity would require knowledge of survival and reproduction in captivity – if it were as good as in the wild, there would already be at least 3 extant captive populations. So, vital rates should probably differ for captive fish, or catastrophes occurred in the captive populations that are not included in the model. This is not considered – and could only be done by scenario exploration.

- I distrust the results of assessments from elasticity analysis because it requires the assumption of a stable age distribution – is there any evidence that the DHP age distribution is stable? What are the expected population changes as you perturb each vital rate, given the known or suspected age distributions? I think the other sensitivity parameters estimated are sufficient – although there could be additional runs with different age structures.
- Line 36 – the first decline was caused by a human-related accident that killed about one-third of the fish (Manning & Wullschleger, 2004, Proceedings of the Desert Fishes Council 36:39)
- Lines 49-51 – It would be great to have citations for this statement.
- Line 100 – This is a confusing point across papers in the literature – what is referred to here as ‘intrinsic’ rate of increase is actually ‘observed’ rate of increase. With density-independent growth, they are the same, but for density-dependent growth they are not. If the word ‘intrinsic’ is dropped, I think the problem is solved.
- I’m not sure equations 4 and 5 need to be presented, just referenced.
- Lines 164-168 – I like this approach – it forces thinking about the results.
- Lines 249-251 – this is a surprise – does it make biological sense?
- Lines 316-317 – this is an offhand comment that might be true – is there any advice on how one would determine this in a fish?
- Line 319 – Is the condor situation sufficiently known to the readership that there is no need to provide a sentence of context?
- Line 328 – is there a reason you did not statistically look for a threshold? It could be that there is no need – if you can’t see it, it’s not biologically important.

---

## Round 0.2 · accepted · Accept

· Academic Editor

Accept

I very much enjoyed this paper and mentioned to the editorial staff about your offer of a video and images.